computational chemistry/computer modelling and simulation/chemical engineering

lithium production, electrolysis efficiency, secondary reaction, multiphysical simulation

**Author for correspondence:**
Cheng-Lin Liu
e-mail: liuchenglin@ecust.edu.cn

This article has been edited by the Royal Society of Chemistry, including the commissioning, peer review process and editorial aspects up to the point of acceptance.

# Analysing and optimizing the electrolysis efficiency of a lithium cell based on the electrochemical and multiphase model

Qian-Wen Zhao[1], Cheng-Lin Liu[1,2], Ze Sun[1,2] and Jian-Guo Yu[1,2]

[1]National Engineering Research Center for Integrated Utilization of Salt Lake Resource, and [2]Resource Process Engineering Research Center for Ministry of Education, East China University of Science and Technology, Shanghai, China

C-LL, 0000-0002-6850-4606; ZS, 0000-0002-1436-9832

Based on an electrochemical multiphysical simulation, a method for analysing electrolysis efficiency has been presented that considers the energy consumption required to produce a single kilogram of lithium and for the production of lithium, rather than the voltage in various parts. By adopting them as the criteria for analysing electrolysis efficiency in the lithium cell, several structural parameters have been optimized, such as the anode radius and anode–cathode distance. These parameters strongly affect the cell voltage and the velocity field distribution, which has a significant impact on the concentration distribution. By integrating the concentration distribution, the lithium production and energy consumption per kilogram, lithium is computed. By appointing the minimum of the chlorine and lithium concentration as the secondary reaction intensity, it is clear where the secondary reaction intensity is strong in the cell. The structure of a lithium electrolysis cell has been optimized by applying an orthogonal design approach, with the energy consumption notably decreasing from 35.0 to 28.3 kWh (kg Li)$^{-1}$ and the lithium production successfully increasing by 0.17 mol.

## 1. Introduction

As lithium is the lightest metal, it is widely used in various industrial applications, such as in alloys for aircraft, electrodes for batteries, the pharmaceutical industry and ceramic composition [1]. With the increasing attention being paid to the new energy

sources, the demand for lithium in energy storage is seeing rapid growth, making it the most popular metal in the twenty-first century.

Traditionally, metallic lithium is mainly produced via two technologies: the vacuum reduction method and the molten salt electrolysis method. For the first method, Kroll & Schlechten [2] and Smeets & Fray [3] have adapted silicon, aluminium and magnesium to reduce lithium oxidation and its ore at 1000°C. For the second method, metallic lithium is typically produced by the electrolysis of LiCl, while the raw material results from the ore or brine [4]. The molten salt LiCl–KCl (42 : 58 in mass ratio) with a low eutectic point of 625.15 K has been adapted for electrolysis [5]. These phenomena at each electrode are running according to the following electrochemical reactions:

$$\text{cathode: Li}^+ + \text{e}^- \rightarrow \text{Li}_{(\text{liquid})},$$

$$\text{anode: 2Cl}^- - 2\text{e}^- \rightarrow \text{Cl}_{2(\text{gas})}$$

and

$$\text{total: 2LiCl} \rightarrow 2\text{Li}_{(\text{liquid})} + \text{Cl}_{2(\text{gas})}.$$

Low-resistance graphite works as the anode and the steel is the cathode. The electrolysis method is operated at 693.15 K with the purity of lithium being 99%. This method is more mature and stable, which means it is widely applied in industrial plants. And many studies have been conducted to improve the electrolysis process. However, the energy consumption for electrolysis is high, and the electrolysis efficiency is low. It is important to optimize the lithium electrolysis cell to reduce the cost of lithium production and to meet the increasing demand for lithium. The optimization of various parameters in the lithium cell, for example, the depth of electrolyte, anode–cathode distance (ACD) and electrode height, is significant for electrolysis efficiency and saving energy in such an energy-intensive industrial process. However, the conditions inside the electrolysis cell, such as a temperature over 693.15 K and lack of space, make the study unobservable and dangerous. The phenomena inside the electrolysis cell include mass transfer, momentum transfer, heat transfer and reaction on electrodes. The interaction of these phenomena and the harsh electrolysis conditions decrease the feasibility of the experimental study and hinder the discovery of insights on the optimization of the electrolysis cell. With the development of computing capability, mathematical modelling has become available and is effective for investigating the mechanisms involved in the electrolysis process. This simulation approach makes it possible to investigate the phenomena under various factors safely and economically.

Several studies [6–11] have been conducted on simulating the electrolysis phenomena in cells, such as electrolyte flowing and bubble elevating. Vogt [12,13] studied the gas-evolving phenomenon and found that it strengthens mass transfer. For thermoelectric coupling, some studies [14–16] found that current intensity and electrolyte height are important for the thermal balance in a cell, which considerably affects the electrolysis efficiency and production. In the aluminium reduction cell, Tessier *et al.* [17] developed a multiblock partial least-squares modelling approach for multivariate analysis and monitoring of aluminium reduction smelters and other electrochemical processes, and Zhang *et al.* [18] developed a microscale modelling approach for the investigation of bubble dynamics in the aluminium smelting process. Zhan *et al.* [19] used the a three-dimensional computational fluid dynamics - population balance model (CFD PBM) to analyse the effects of different cell designs and operating parameters on the gas–liquid two-phase flows and bubble distribution characteristics under the anode bottom regions. Vukasin *et al.* [20] investigated the fluorine electrolysis cell by coupling electric currents, heat transfer, diluted species transport and two-phase flow. However, few studies [20,21] focused on studying the electrolysis process while coupling mass-electric-concentration fields and reaction on electrodes. The simulation and optimization of the cell will be more accurate if the above fields and reaction are taken into consideration. Several researchers have optimized the lithium electrolysis cell based on the effect of structural and operational parameters on the electric field to improve the electrolysis efficiency and reduce energy consumption. However, no work has been done to optimize the electrolysis cell by considering the secondary reaction in cell, which is one of the critical reasons affecting the electrolysis efficiency. Reports on the simulation of lithium electrolysis are few [22–24].

Based on the above discussion, this study aims to optimize the structural parameters by taking the above fields and reaction into consideration to accurately simulate and optimize the cell. A mathematical model, with electric-concentration-flow fields and reactions, has been built to investigate the effect of various parameters on electrolysis efficiency, which is characterized by the lithium production and the energy consumption per kilogram of lithium in the cell. By analysing the concentration of products in the cell, this study considers reducing the secondary reaction as a critical step to improve the electrolysis efficiency. Finally, an energy-saving and economical lithium

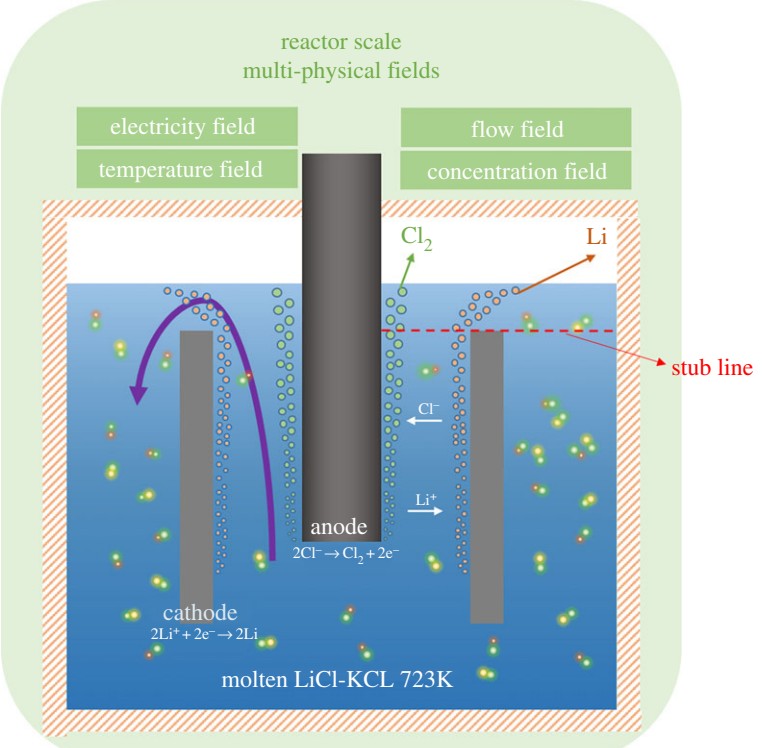

**Figure 1.** Phenomena in the lithium electrolysis cell.

electrolysis cell will be developed by analysing the lithium production and the secondary reaction occurring between the metal lithium droplet and chlorine bubble.

# 2. Model and method

## 2.1. Simulation methodology

The lithium electrolysis process involves direct current, mass transfer and electrolyte flow. The interaction among those phenomena and the harsh electrolysis condition makes the laboratory-based research difficult. In this study, the FEM software COMSOL Multiphysics is applied to simulate the electrolysis process by coupling the velocity field, electric field, concentration field and reactions on electrodes.

The core of electrolysis is the cell and the operational conditions. Important components in the electrolysis cell consist of the graphite bar as the anode, the steel flat as the cathode and the diaphragm in some cells. The most popular commercial electrolysis cell is diaphragmless and is chosen for investigation in this study. The configuration of the industry electrolysis cell is shown in figure 1. For the purpose of improving the computational efficiency and reducing computation time, the cell is simplified into a two-dimensional axisymmetric model. A stub line, which is on the cathode, is created to analyse the velocity and concentration distribution.

The binary molecular diffusion coefficients for $Li^+$, $K^+$ and $Cl^-$ have been measured by Janz & Bansal [25] at high temperatures. According to Oliaii *et al*. [26], the accuracy of these diffusion coefficients is validated by comparing the electrolyte conductivity—estimated below using these diffusion coefficients and initial ion concentrations—with that of Van Artsdalen & Yaffe ($157\,S\,m^{-1}$) [27]. Table 1 lists the properties of the reactants and the products estimated at 723.15 K.

## 2.2. Mathematical model

### 2.2.1. Electric field

During the electrolysis process, a direct current of 5 kA has been added on the anode and it flows towards the cathode through the electrolyte. While the electric field is the prerequisite of the

**Table 1.** Properties of the reactants and products.

| name | density (kg m$^{-3}$) | diffusion coefficiency (m$^2$ s$^{-1}$) | electrical conduction (S m$^{-1}$) | viscosity (μPa s) |
|---|---|---|---|---|
| LiCl–KCl(l) | 1648 | * | 218 | 1590 |
| Cl$^-$ | * | $3.0 \times 10^{-9}$ | * | * |
| Li$^+$ | * | $2.0 \times 10^{-9}$ | * | * |
| Cl$_2$ | 1.77 | $3.6 \times 10^{-9}$ | 0 | 27 |
| Li(l) | 512 | $3.7 \times 10^{-9}$ | 84.7 | 4610 |

*Not needed in this model.

electrolysis reaction, it is also the heat source for the whole cell. The electric potential will contribute to the migration of ions and affect the thermal distribution by producing heat during the electrochemical reaction on the electrodes.

In this instance, the electric field could be described by Ohm's law, continuity law and Gauss's law as given below

$$J = \sigma E, \tag{2.1}$$

$$\frac{\partial q}{\partial t} + \nabla \cdot J = 0 \tag{2.2}$$

and

$$E = -\nabla\varphi, \tag{2.3}$$

where $J$ (A m$^{-2}$) is the current density; $\sigma$ (S m$^{-1}$) is the electric conductivity; $E$ (V m$^{-1}$) is the electric field; $q$ (C m$^{-3}$) represents the charge density; and $\varphi$ (V) is the electric potential.

Connecting the above equations, we obtain the following:

$$\nabla \cdot (\sigma\nabla\varphi) = \frac{\partial q}{\partial t}. \tag{2.4}$$

During the electrolysis process in the cell, the electric field stays stable and there is no other charge produced. Equation (2.4) can be simplified as

$$\nabla \cdot (\sigma\nabla\varphi) = 0. \tag{2.5}$$

The electrodes are electric-conductive with an appointed boundary condition, while the other boundaries are insulated.

The effect of bubbles on the conductivity of the electrolyte is considered for the relationship between the electrical resistivity of the electrolyte and gas volume fraction.

### 2.2.2. Velocity field

The electrolysis cell includes three phases: the molten salt mixture of LiCl–KCl as electrolyte, the liquid lithium produced from the vertical cathode and the gaseous chlorine evolving from the vertical anode. The effect of the lithium on the velocity field compared to the chlorine bubble is too small to take into consideration.

According to the features of low gas concentration and the gas–liquid phases flowing along vertical electrodes, the turbulent bubble flow model is applied to simulate the velocity and normal $k$–$\varepsilon$ is chosen to depict the turbulence state of the flow [28].

$\theta_l$ and $\theta_g$ represent the volume fraction of the liquid phase and the gas phase, respectively. The letters 'l' and 'g' represent the liquid and the gas phase, respectively. As there is no mass transfer in the two phases, the continuous equation can be rewritten as

$$\frac{\partial(\theta_k\rho_k)}{\partial t} + \nabla \cdot (\theta_k\rho_k u_k) = 0, \quad k = l, g \tag{2.6}$$

and

$$\theta_l = 1 - \theta_g. \tag{2.7}$$

The normal $k$–$\varepsilon$ turbulence model is derived from the Navier–Stokes equation. The turbulent kinetic energy $k$ (m$^2$ s$^{-2}$) and turbulent dissipation rate $\varepsilon$ (m s$^{-3}$) solved by the normal $k$–$\varepsilon$ turbulence model are

$$\rho_l \frac{\partial k}{\partial t} + \rho_l u_l \cdot \nabla k = \nabla \cdot [\mu_{\text{eff},k} \nabla k] + P - \rho_l \varepsilon + S_k \tag{2.8}$$

and

$$\rho_l \frac{\partial \varepsilon}{\partial t} + \rho_l u_l \nabla \varepsilon = \nabla \cdot \left[ \mu_l + \frac{\mu_T}{\sigma_\varepsilon} \nabla \varepsilon \right] + C_{\varepsilon 1} \frac{\varepsilon}{k} P + C_\varepsilon S_k \frac{\varepsilon}{k} - C_{\varepsilon 2} \rho_l \frac{\varepsilon^2}{k}. \tag{2.9}$$

The inducement source $S_k$ is the turbulence force induced by bubbles, and can be expressed by the following equation:

$$S_k = -C_k \theta_g \nabla p \cdot (u_l - u_g), \quad C_k = 1, \tag{2.10}$$

where

$$P = \mu_T \left\{ \nabla u_l : [\nabla u_l + (\nabla u_l)^{\mathrm{T}}] - \frac{2}{3} (\nabla \cdot u_l)^2 \right\} - \frac{2}{3} \rho_l k \nabla \cdot u_l. \tag{2.11}$$

In this model, $C_\varepsilon = 1.92$, $C_{\varepsilon 1} = 1.42$, $C_{\varepsilon 2} = 1.68$, $\sigma_\varepsilon = 1.3$ [29].

In this equation, $u$ is the speed (m s$^{-1}$), $p$ represents the pressure (Pa), $\rho$ is the fluid density (kg m$^{-3}$), $g$ is the gravitational acceleration (m s$^{-2}$) and $\mu_{\text{eff},k}$ is the effective viscosity. For the liquid phase, it is the sum of the dynamic viscosity $\mu_l$ and the turbulence viscosity

$$\mu_{\text{eff}} = \mu_l + \mu_T. \tag{2.12}$$

In this equation, the turbulence viscosity is

$$\mu_T = \rho_l C_\mu \frac{k^2}{\varepsilon}, \quad C_\mu = 0.0845. \tag{2.13}$$

Compared to the turbulence viscous stress, the effect of molecule viscous stress on multiphase is too small to consider. The momentum equations become

$$\frac{\partial (\theta_k \rho_k u_k)}{\partial t} + \nabla \cdot (\theta_k \rho_k u_k u_l) = -u_k \nabla p + \nabla \cdot (\theta_k T_k^{\text{turb}}) + \theta_k \rho_k g + F_{\text{added}}, \quad k = l, g \tag{2.14}$$

and

$$T_k^{\text{turb}} = -\mu_{\text{eff},k} \left( \nabla u_k + (\nabla u_k)^{\mathrm{T}} - \frac{2}{3} I (\nabla u_k)^{\mathrm{T}} \right). \tag{2.15}$$

In this equation, $F_{\text{added}}$ is the added force on the fluid (N m$^{-3}$) and $T_k^{\text{turb}}$ is the turbulence stress (N m$^{-3}$).

### 2.2.3. Concentration field

In this model, only reacting ionic species and product species have been included. In reality, the electrolyte in the lithium electrolysis cell is a concentrated solution. However, for the product species as the main factor for the electrolysis efficiency, it is a diluted solution. Hence, a diluted solution is considered to solve the concentration field.

There are three mechanisms for the transport of the ionic species inside the lithium electrolysis cell: convection, diffusion and migration. Considering those mechanisms by means of the Nernst–Planck equation, the flux $N_i$ of each $i$ in the electrolysis cell can be written as

$$N_i = c_i V - D_{i,\text{eff}} \nabla c_i - z_i F u_{m,i} c_i \nabla \varphi. \tag{2.16}$$

In this equation, $u_{m,i}$ represents the ions mobility, which can be calculated by the Nernst–Einstein equation

$$u_{m,i} = \frac{D_{i,\text{eff}}}{RT}. \tag{2.17}$$

**Table 2.** Initial conditions in the model.

| transfer process | description |
| --- | --- |
| electrical potential | cell voltage equals 0 V |
| mass transfer | reactive species concentration equals 19.2 kmol m$^{-3}$ |
| velocity field | velocity equals zero |

**Table 3.** Boundary conditions in this model.

| boundary | velocity field | electric field | concentration field |
| --- | --- | --- | --- |
| anode (contact with electrolyte) | gas inlet, slip for liquid | 5 kA | $2Cl^- - 2e^- \rightarrow Cl_{2(gas)}$ |
| cathode (contact with electrolyte) | no slip for liquid | 0 V | $Li^+ + e^- \rightarrow Li_{(liquid)}$ |
| outlet | slip for liquid, gas outlet | electrical insulation | gas outlet |
| side-wall | no slip for liquid | electrical insulation | wall |
| bottom | no slip for liquid | electrical insulation | wall |

According to the mass conservation law and the charge conservation law

$$\frac{\partial c_i}{\partial t} = \nabla \cdot N_i + R_i \tag{2.18}$$

and

$$\sum_i z_i c_i = 0, \tag{2.19}$$

where $R_i$ means the homogeneous reaction rate of species $i$ in the electrolyte. In the cell, $R_i$ is equal to the electrolysis reaction generation rate when on the electrodes and equals zero. $z_i$ represents the electric charge of each particle. To consider the eddy diffusivity on ions transfer, $D_{i,\text{eff}}$ equals the sum of the effective molecular diffusion coefficient and the turbulent diffusion coefficient. For most practical cases, the turbulent Schmidt number, defined as eddy viscosity/eddy diffusivity, is assumed to be 1. Consequently, the turbulent diffusion coefficient is computed according to the turbulent kinematic viscosity from the velocity field. In this instance, $\varphi$ is the electrical potential, which is obtained from the general electrolyte current conservation law

$$\nabla \cdot \left( -F \left( \nabla \sum_I z_i D_{i,\text{eff}} c_i \right) - \sigma_l \nabla \varphi_l \right) = 0, \tag{2.20}$$

where

$$\sigma_l = F^2 \sum_i z_i^2 u_{m,i} c_i. \tag{2.21}$$

For the electroactive ions on the electrode surfaces, the flux is related to the current through Faraday's law.

## 2.3. Initial and boundary conditions

The initial conditions for the electrolysis cell are listed in table 2.

The boundary conditions used in this model are listed in table 3.

This article is aimed at analysing and optimizing the structure of the electrolysis. The boundary condition is controllable in the routine operation process. So, in this article the boundary is stable in this mathematical model and just the structure parameters change to get better electrolysis efficiency.

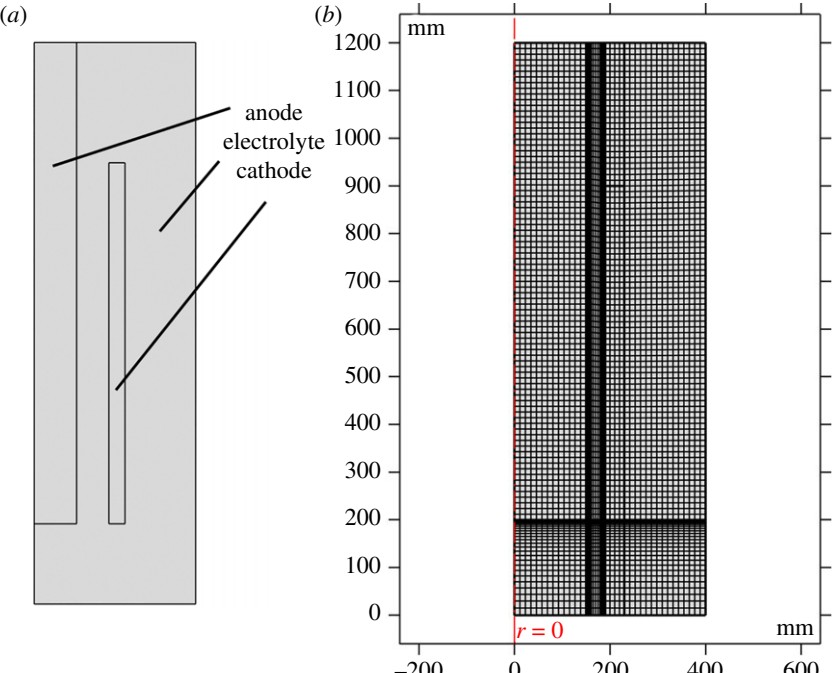

**Figure 2.** (*a*) Structure and (*b*) mesh of the model.

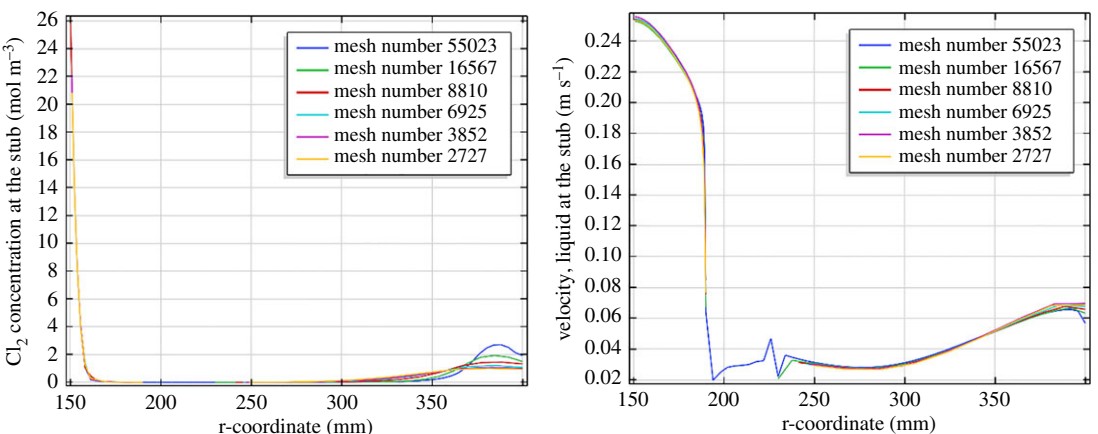

**Figure 3.** Concentration of $Cl_2$ and velocity distribution along the stub with the various elements.

## 2.4. Mesh independency

The mesh independency aims to ensure that the quality of the simulation results would not be influenced by the number of the mesh size. Structured mesh has been tested with different numbers between 2720 and 55 023. As shown in figure 2, finer meshes are used near the electrodes, which have the maximum gradient of concentration and velocity. At the same time, the stub line in figure 1 with the largest gradients of the concentration and velocity is chosen as an analysis line.

Figure 3 shows the $Cl_2$ concentration distribution and liquid velocity distribution, which show a close value at the stub with different mesh elements. The main result for different fields such as the $Cl_2$ concentration are mesh-independent when the elements exceed 2727. The following model will adopt 6925 elements.

## 2.5. Validation of simulation model

Owing to the harsh conditions prevailing inside molten salt electrochemical cells, the experimental data are difficult to measure. A mathematical model with a ratio of 1 : 1 to the industrial cell is established to simulate the electrolysis process.

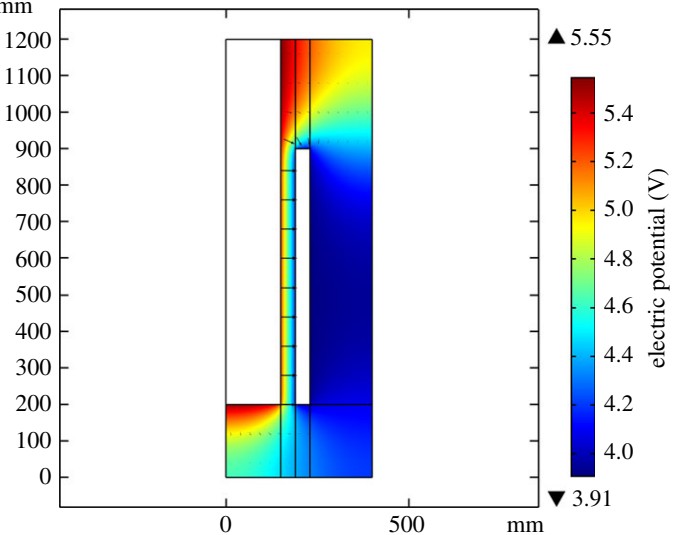

**Figure 4.** Electric potential (V) and electric field arrow diagram.

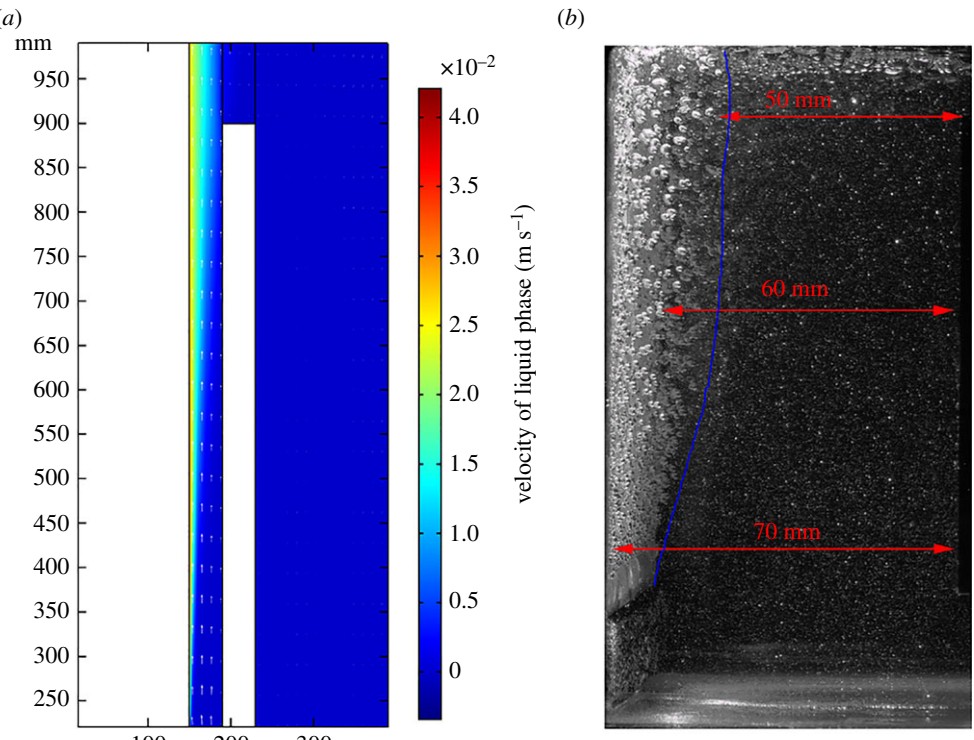

**Figure 5.** (*a*) Velocity of liquid phase (m s$^{-1}$). (*b*) Liquid flowing arrow diagram.

According to the simulation result, after 40 s, these fields arrive at steady state. Figure 4 shows the electrolyte potential is 1.64 V while fixing the current at the industrial value of 5 kA. Ignoring the external voltage drops and the evolution of the cell voltage over time, and adding the reaction equilibrium potential of 3.72 V and the over-potential of 0.18 V, the cell electrolyte potential equals 5.55 V. Taking the electrodes' voltage of 0.68 V into consideration, the sum of cell potential equals 6.22 V, which is located in the range of 6.00–7.00 V specified by the industry. As shown in figure 5*a,b*, as gas bubbles elevate because of buoyancy, the liquid phase flows to the outlet because of the drag force. The bubble layer of gas distribution grows thicker from 10 to 20 mm, which agrees with the experiment data obtained by Liu *et al.* [30].

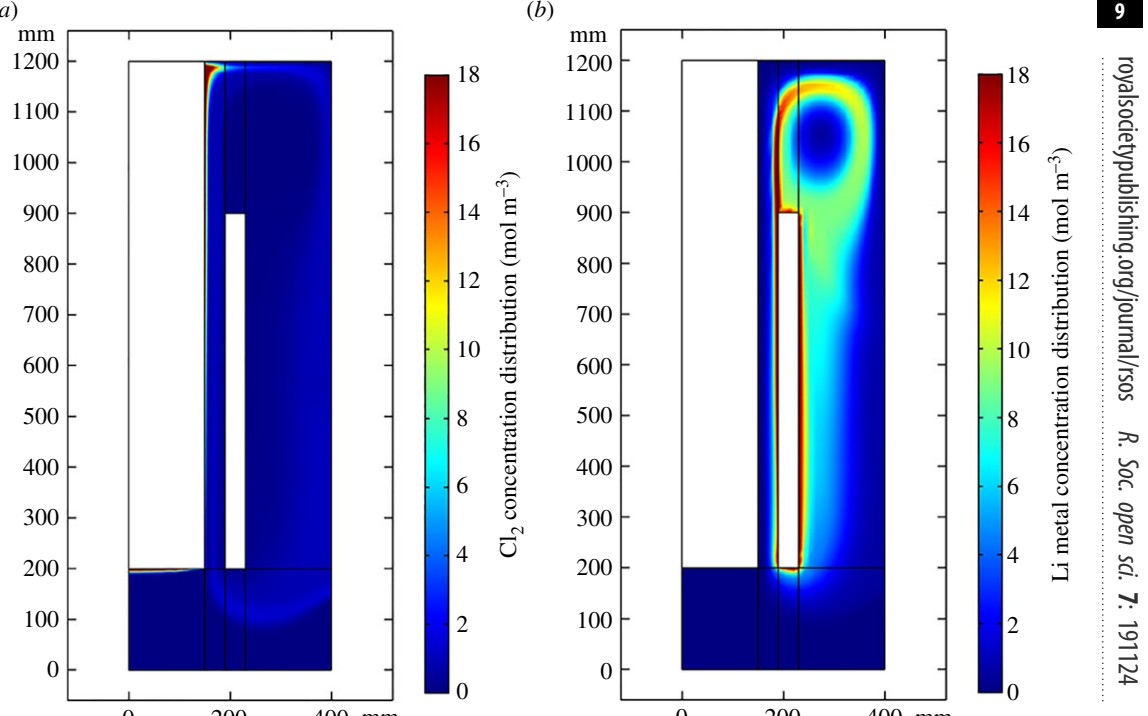

**Figure 6.** (a) $Cl_2$ Concentration distribution (mol m$^{-3}$) and liquid flowing arrow diagram. (b) Li metal concentration distribution (mol m$^{-3}$).

Comparing the energy consumption per kilogram lithium from this simulation model of 35 kWh (kg Li)$^{-1}$ to the value obtained from industrial cell and experiment data [31] of 25–50 kWh (kg Li)$^{-1}$, the agreement between the energy density obtained from industry and simulation is considered reasonable.

# 3. Results and discussion

## 3.1. Analysing method of electrolysis efficiency based on concentration field

In this study, a method for analysing the electrolysis efficiency has been promoted by coupling electric-concentration-velocity fields and reactions. According to previous studies [24,32], most of them account for the electrolysis efficiency by analysing the electric potential. However, while the electric potential increases, the objective lithium production also increases. Energy consumption per kilogram of lithium may decrease or increase. Under this condition, it is not accurate to appoint the electric potential as the indicator of whether the electrolysis cell is energy-saving or economical. As a result, energy consumption per kilogram of lithium and the lithium production have been adopted as the criteria to optimize the lithium electrolysis cell.

Figure 6 shows that the distributions of $Cl_2$ and liquid metal lithium follow the velocity field. However, there are few electrolysis models that take the product concentration into consideration. This model, which is coupled with multiphysical fields and the reaction, analyses not only the product concentration in the cell but also the secondary reaction intensity.

The electric reaction relating to current on the electrode surface is as follows:

$$\sum_{ox} v_{ox}S_{ox} + ne \leftrightarrow \sum_{red} v_{red}S_{red}. \qquad (3.1)$$

The reactant $S_{ox}$ on this electrode–electrolyte interface gets electrons and is reduced to the product $S_{red}$. $v_{ox}$ is the stoichiometric number. Those species are the only mass flowing out the reaction surface. The species flow can be calculated by Faraday Law

$$R_i = \sum_m \frac{v_{i,m}\dot{i_1}}{nF}, \qquad (3.2)$$

where $i_1$ is the local current density (A m$^{-2}$) and $n$ is the reacting electron number.

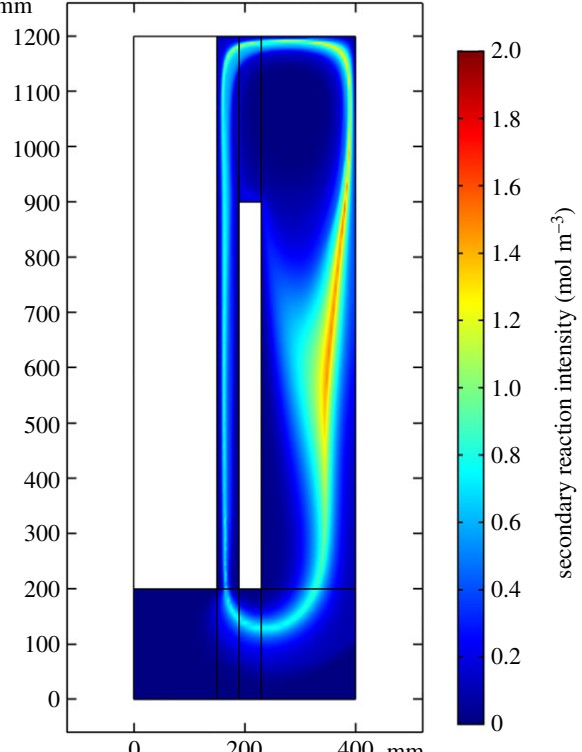

**Figure 7.** Secondary reaction Intensity (mol m$^{-3}$).

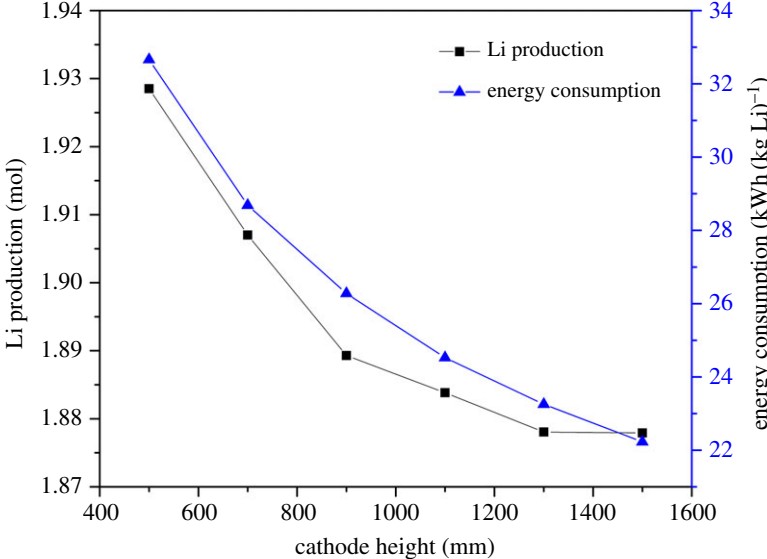

**Figure 8.** Effect of cathode height on electrolysis efficiency.

Both $Cl_2$ and lithium are partly brought to the region between the cathode and wall. This contributes to the secondary reaction and brings down the electrolysis efficiency. The distribution of the secondary reaction mainly follows the lithium distribution because $Cl_2$ diffuses considerably more quickly than lithium.

This research presents not only the product concentration distribution in the entire cell but also the secondary reaction intensity distribution, as shown in figure 7. By this means, it can be easy to illustrate how the product distribution is affected by various factors. Then, optimization measures can be taken to avoid the secondary reaction. The total lithium production and the

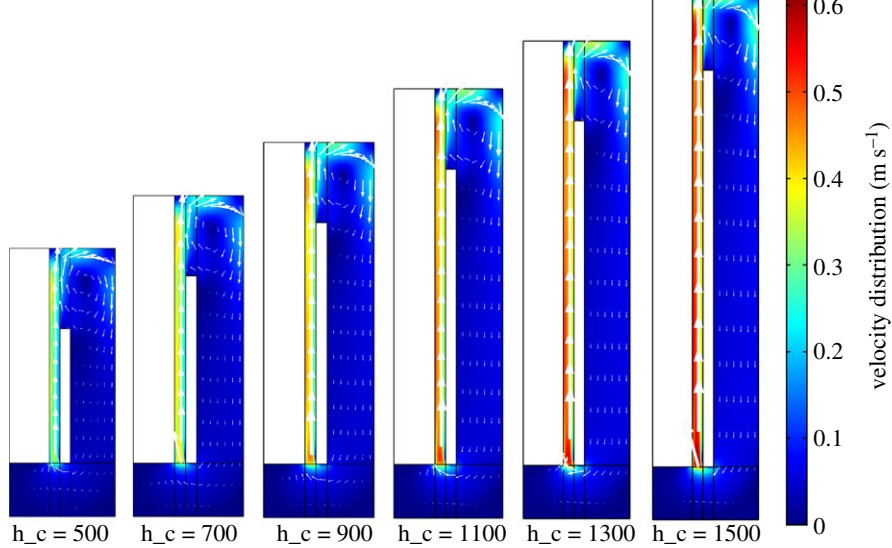

**Figure 9.** Velocity field along various cathode heights velocity distribution (m s$^{-1}$).

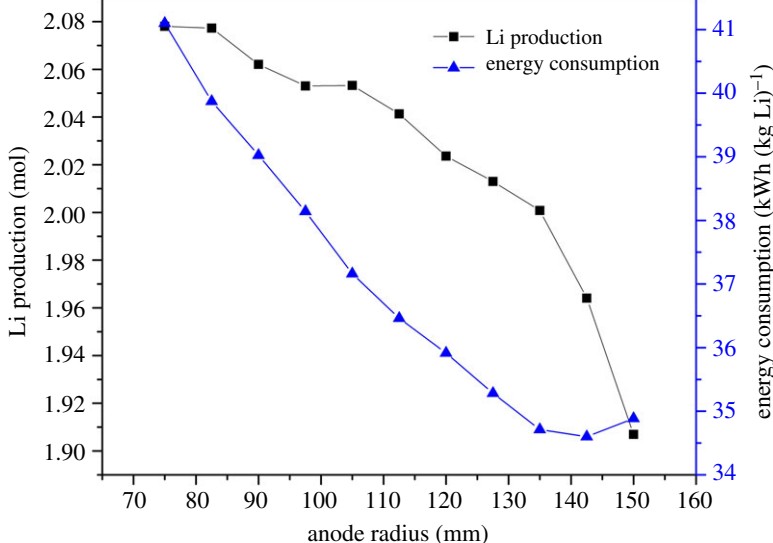

**Figure 10.** Effect of anode radius on electrolysis efficiency.

secondary reaction are calculated by volume integrating. Finally, the electrolytic consumption per kilogram of lithium and total lithium production are both considered to optimize the lithium electrolysis cell.

## 3.2. Effect of cathode height on electrolysis efficiency

Figure 8 shows that the metal lithium production and the energy consumption decrease quickly, as logarithmic functions, as cathode height increases from 400 to 900 mm. While the cathode height becomes higher, both the chlorine bubble and the lithium go through a longer distance towards the stub line, which gives the gas and the liquid phase time to develop turbulence, strengthening the mass transfer, as shown in figure 9. As a result, the secondary reaction increases sharply and the lithium production decreases consequently. As the available electrolysis area on the electrodes increases, the energy consumption decreases because the current density and electric potential decrease faster. It also tells us that the available electrolysis area has a significant influence on the electric potential. When the height increases above 900 mm, both phases have already arrived at the turbulent state, and the secondary reaction does not aggravate along the increasing velocity field.

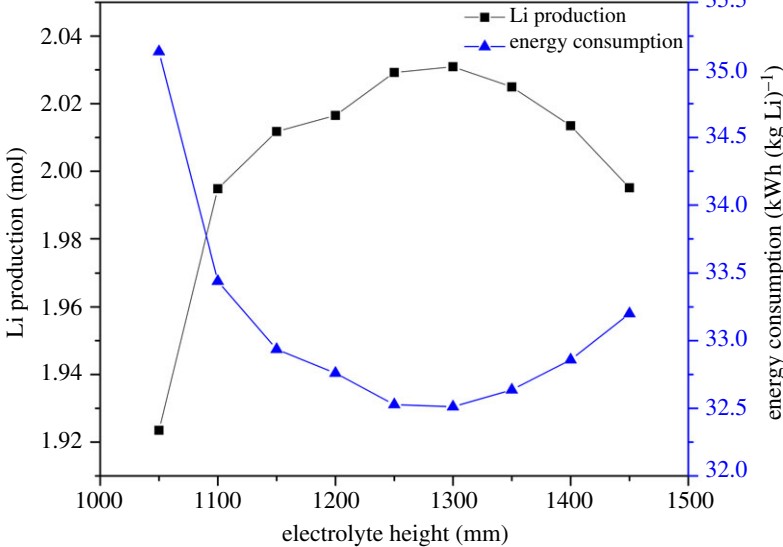

**Figure 11.** Effect of electrolyte height on electrolysis efficiency.

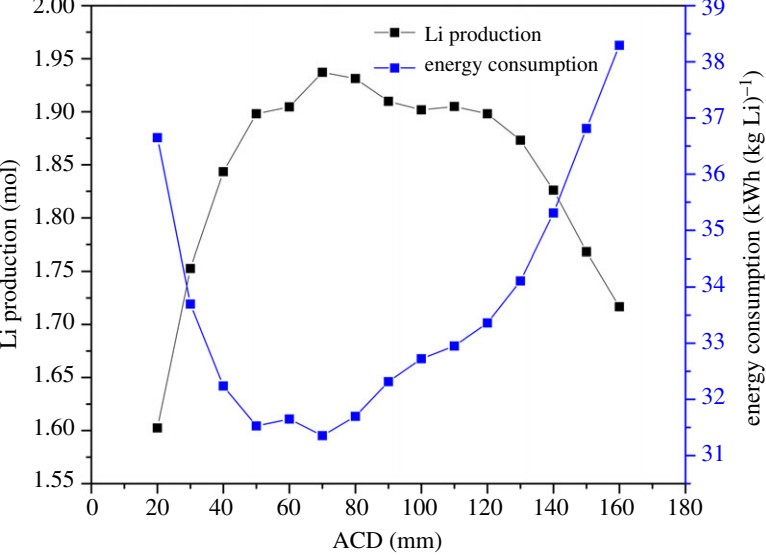

**Figure 12.** Effect of ACD on electrolysis efficiency.

The electrode surface on the upper level is blocked by the gas phase, the current density on the electrode surface keeps steady and so does the electric potential. Therefore, the energy consumption decreases along the increasing cathode height. To get high lithium production, it is more economical and efficient to keep the cathode height lower than 900 mm. As a result, 700 mm has been recommended as the most efficient cathode height.

## 3.3. Effect of anode radius on electrolysis efficiency

The anode is made of graphite with low resistance and will be corroded by the electrolyte. Figure 10 shows as the radius reduces, the ACD increases, the gas phase will depart from the metal lithium phase, the secondary reaction—$Cl_2$ reacting with metal lithium—decreases; therefore, the metal lithium production increases. The energy consumption per kilogram of lithium increases from 34.5 to 41.2 kWh (kg Li)$^{-1}$ when the anode radius decreases from 150 to 75 mm. It can be concluded that the anode radius has a stronger effect on the electric potential than on the lithium production. Furthermore, the slope of the energy consumption line grows along the anode radius. However, the contrary results for the lithium production are presented.

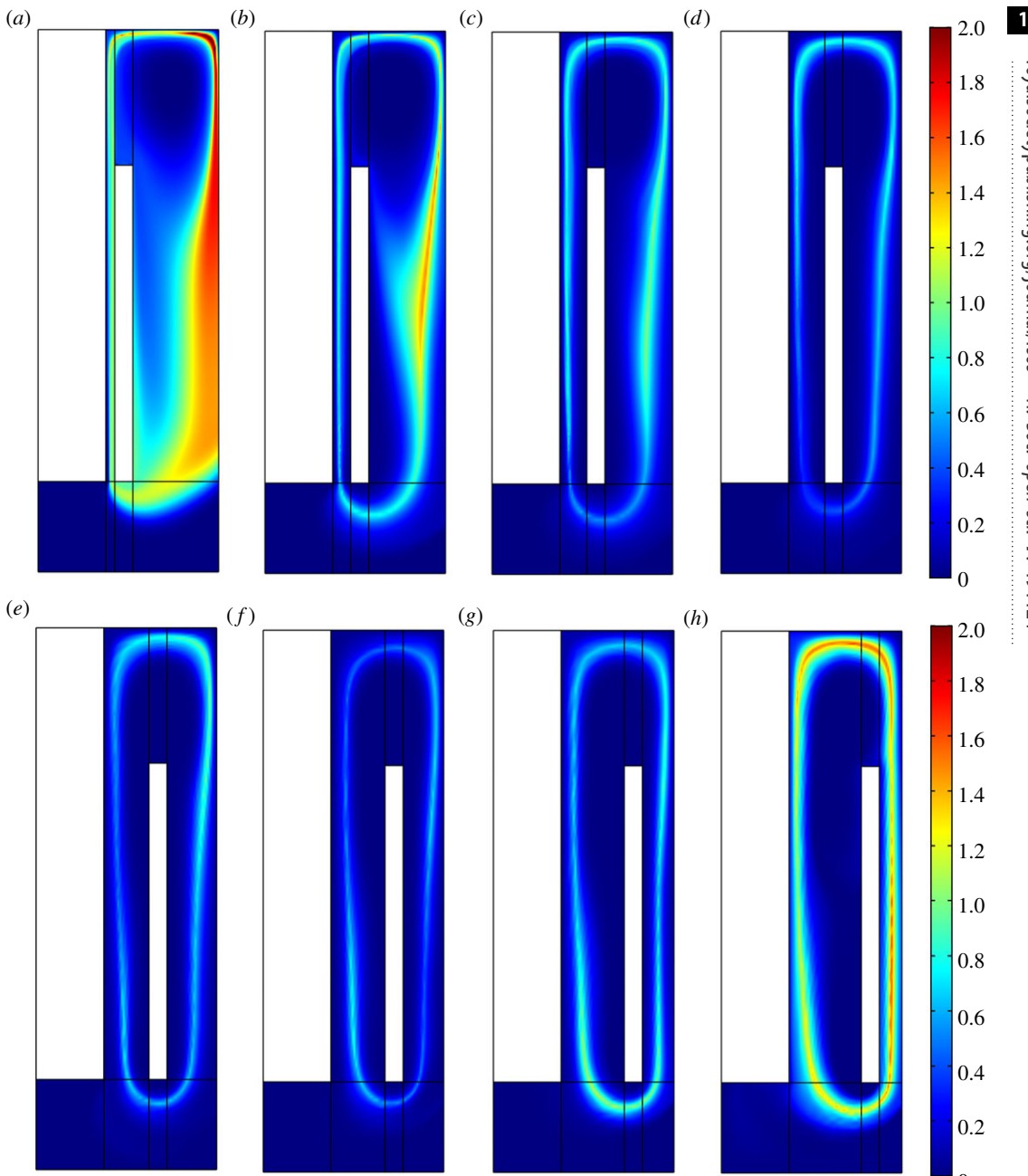

**Figure 13.** (a–h) Secondary reaction intensity with various ACD.

## 3.4. Effect of electrolyte height on electrolysis efficiency

As the electrolyte level increases from 1050 to 1300 mm, the lithium production grows sharply and then decreases from 1300 to 1450 mm, as shown in figure 11. As the electrolyte level has no influence on the total energy consumption, the energy consumption changes along the lithium production. As the available flow passageway above the cathode grows from 150 to 350 mm, the lithium departs from the chlorine bubbles, the secondary reaction decreases obviously. However, while the electrolyte level continues to increase, the lithium production increases slowly and begins to decrease because the wider passageway above the cathode makes no more influence on the lithium production. However, after 1300 mm, deeper electrolyte height requires more time for the bubble flowing from the bottom to the surface, which causes the secondary reaction to increase much more the effect of the wider passageway. Therefore, the lithium production decreases. The lithium production stays high, while the electrolyte height increases from 1200 to 1350 mm.

**Table 4.** Orthogonal design and simulation results for lithium electrolysis cell.

| case no. | ACD (mm) | cathode height (mm) | anode radius (mm) | Li production (mol) | Energy consumption (kWh (kg Li)$^{-1}$) |
|---|---|---|---|---|---|
| 1 | 80 | 500 | 75 | 2.08 | 28.32 |
| 2 | 80 | 700 | 90 | 2.04 | 39.11 |
| 3 | 80 | 900 | 105 | 2.04 | 33.57 |
| 4 | 80 | 1100 | 120 | 2.04 | 29.80 |
| 5 | 80 | 1300 | 135 | 1.99 | 27.95 |
| 6 | 90 | 500 | 90 | 2.11 | 44.36 |
| 7 | 90 | 700 | 105 | 2.04 | 37.56 |
| 8 | 90 | 900 | 120 | 2.01 | 32.61 |
| 9 | 90 | 1100 | 135 | 1.98 | 30.02 |
| 10 | 90 | 1300 | 75 | 1.97 | 32.50 |
| 11 | 100 | 500 | 105 | 2.05 | 43.83 |
| 12 | 100 | 700 | 120 | 2.06 | 36.05 |
| 13 | 100 | 900 | 135 | 2.03 | 31.94 |
| 14 | 100 | 1100 | 75 | 2.01 | 34.44 |
| 15 | 100 | 1300 | 90 | 2.00 | 30.82 |
| 16 | 110 | 500 | 120 | 2.05 | 42.43 |
| 17 | 110 | 700 | 135 | 2.06 | 35.17 |
| 18 | 110 | 900 | 75 | 2.04 | 37.13 |
| 19 | 110 | 1100 | 90 | 2.02 | 32.88 |
| 20 | 110 | 1300 | 105 | 2.01 | 29.75 |
| 21 | 120 | 500 | 135 | 2.05 | 41.43 |
| 22 | 120 | 700 | 75 | 2.08 | 41.18 |
| 23 | 120 | 900 | 90 | 2.06 | 35.36 |
| 24 | 120 | 1100 | 105 | 2.03 | 31.74 |
| 25 | 120 | 1300 | 120 | 2.01 | 29.12 |

## 3.5. Effect of anode–cathode distance on electrolysis efficiency

Figure 12 shows that for energy consumption, there is a lowest point with a value of 31.4 kWh (kg Li)$^{-1}$ at an ACD of 70 mm, which also has the largest production of metal lithium. The lithium production keeps increasing and energy consumption keeps decreasing, while the ACD varies from 20 to 70 mm. In this range, as shown in figure 13, it is found that the lithium production increases because the secondary reaction reduces, which can avoid the elevation of the chlorine bubble bringing the lithium to the back of the cathode. While the ACD is small, the convection of the chlorine and lithium is strong, the mass transfer is strengthened and the $Cl_2$ tends to diffuse to the cathode surface, where the secondary reaction is intensive. When the ACD is larger than 70 mm, the convection becomes weak and the intensity of the secondary reaction becomes small, so the extent of lithium production remains constant at a high value, but the electric potential increases. As a result, the energy consumption per kilogram lithium increases. After the ACD exceeds 120 mm, the lithium production decreases and the energy consumption increases quickly along the ACD. As the ACD increases, as shown in figure 13, during the above process, the intensity region of the secondary reaction begins to shift from the cathode surface to the region between cathode and wall. The secondary reaction mostly occurs near the junction of the anode and outlet or below the cathode.

As a result, it is recommended that the ACD keeps in the range from 45 to 120 mm, while the energy consumption is low and the lithium production is still high. In conclusion, the ACD has a higher influence on the secondary reaction than on the electric potential when it is below 70 mm; when the

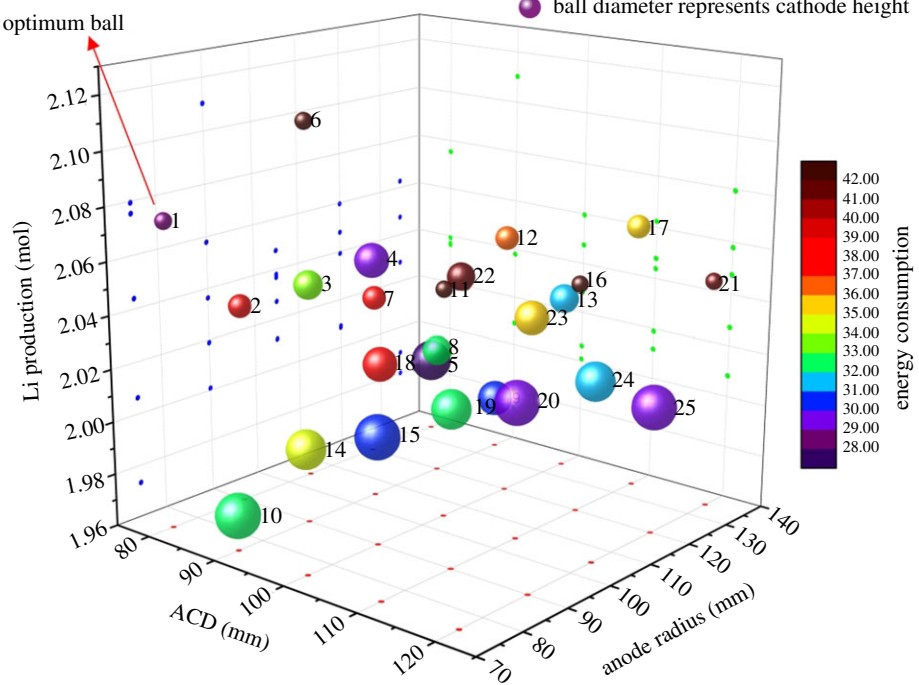

**Figure 14.** Simulation results of lithium electrolysis cell by orthogonal design.

ACD is in the range from 70 to 110 mm, the effect on the secondary reaction is weak and the electric potential keeps increasing; and when the ACD is over 120 mm, it contributes more to the energy consumption than to the lithium production.

## 3.6. Optimization of the 5 kA lithium electrolysis cell

Based on the above discussion, five structural parameters have been chosen to investigate their effect on the lithium production and energy consumption. The distance between the anode and cathode, cathode height and anode radius have considerable influence on the electrolysis efficiency. To find the optimal design conditions for the lithium electrolysis cell, the orthogonal design approach is adopted to find the optimum value among the three factors. For every factor, five levels have been considered, which are around the most efficient point in the single factor testaments. Ignoring the interactions among those factors, the scheme is approached, as presented in table 4, by the orthogonal array L25(5⁶). Therefore, these simulation results can represent the full consideration of various cell structures.

The scatter distributions in figure 14 show the results of the electrolysis simulation of orthogonal designing. In this figure, the $X$-axis represents the ACD, the $Y$-axis represent anode radius and the ball diameter represents the cathode height. The $Z$-axis represents the lithium production of each case and the colour of each ball represents the energy consumption, the colour changing from red to purple means the energy consumption for each is higher. From the scatter distributions, Ball 1 (case 1) in the highest place gets the highest production with the value of 2.08 mol and the energy consumption 28.3 kWh (kg Li)$^{-1}$. According to the economic value of the energy consumption and the lithium production, case 1 is recommended as the optimal condition for the lithium electrolysis cell. Under the condition that the distance between anode and cathode is 80 mm, the height of the cathode is 500 mm and the anode distance is 75 mm, the energy consumption per kilogram of lithium is 28.3 kWh (kg Li)$^{-1}$ and the lithium production is 2.08 mol.

Comparing the images in figure 15a,b, it is found the secondary reaction intensity has been significantly decreased after optimization. Comparing the images $c$ and $b$, the velocity field after optimization is gentler and more uniform and smaller ACD gives the liquid phase larger velocity, which brings the chlorine and lithium to the back of the cathode. Therefore, the secondary reaction is rather intensive here. After optimization, the secondary reaction mainly happens between the

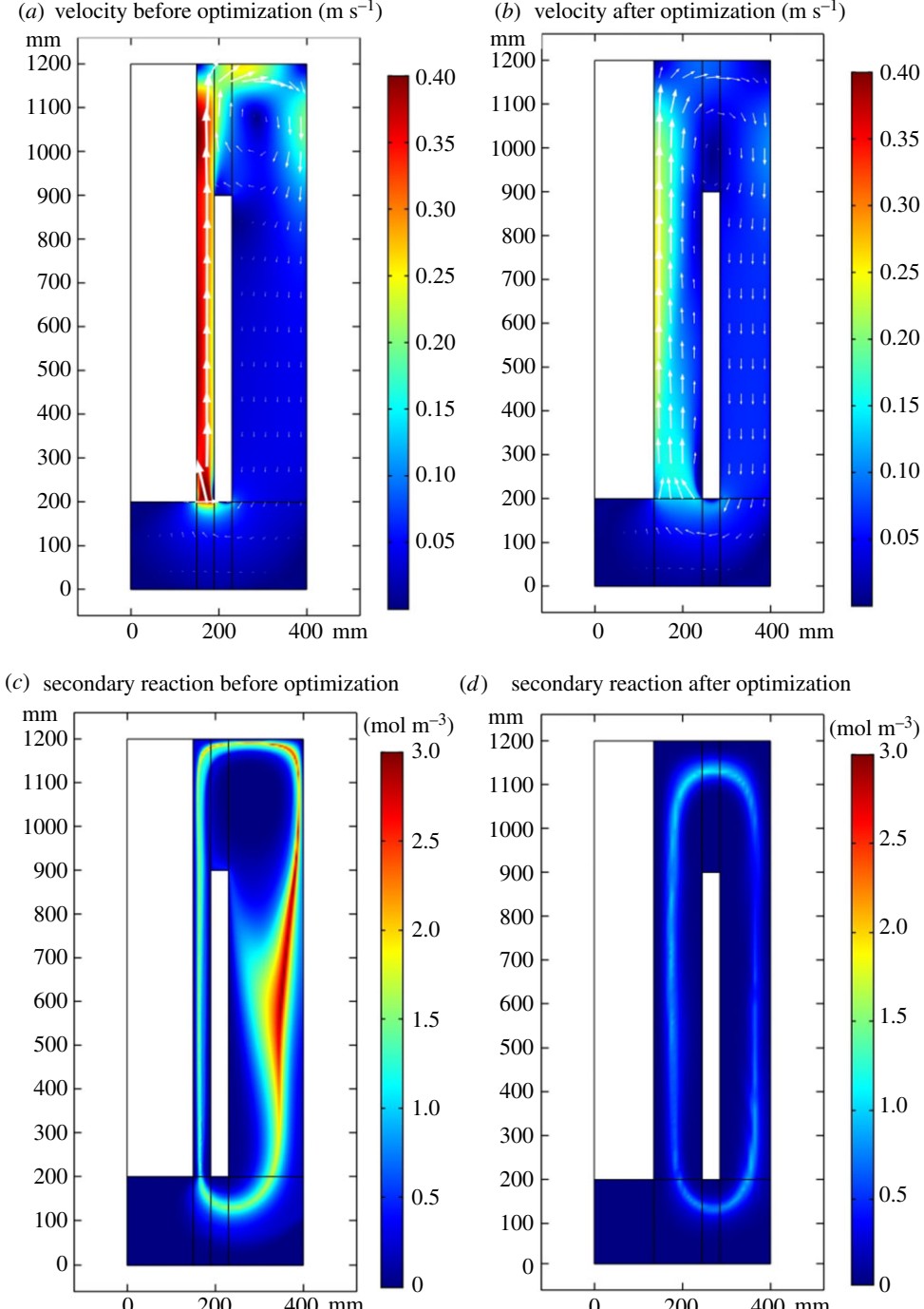

**Figure 15.** (*a*–*d*) Secondary reaction intensity and velocity field before and after optimization.

anode and cathode, and the electrolysis efficiency has considerably improved. After optimization, the velocity distribution will be steadier and the electrolyte will erode the anode and cathode slowly.

## 4. Conclusion

In this research, a model coupled multiphysical fields for simulating lithium electrolysis cell has been developed by applying COMSOL. By simplifying the industrial electrolysis cell on the ratio of 1:1, a two-dimensional axisymmetric cell model has been used to investigate the effect of cell structure parameters on the electrolysis efficiency. A novel method of analysing electrolysis efficiency by considering the

secondary reaction intensity, lithium production and energy consumption for producing each kilogram of lithium was applied. According to the simulation results, the anode radius and ACD have a significant effect on the velocity field, the concentration distribution of metal lithium and chlorine bubble, and the secondary reaction intensity. Owing to higher cathode height and electrolyte height with the larger area of electrolyte covering on the electrodes, the electric potential decreases with higher cathode height and electrolyte height. The influence on the velocity field and concentration distribution is not as obvious as that on the electric potential. As a result, a 5 kA lithium electrolysis cell has been optimized according to the research result. The analysis shows that the secondary reaction intensity has decreased and the energy consumption per kilogram of lithium has decreased to 28.3 from 35.0 kWh (kg Li)$^{-1}$ after optimization, and the lithium production increases by 0.17 mol.

In conclusion, the orthogonal optimization result and FE models coupling the multiphysical fields and reactions presented in this paper are instructive to the optimization of electrolysis cells. Meanwhile, the novel characterization method for analysing the electrolysis efficiency developed in this work throws light on future research on electrolysis.

Data accessibility. Data are available from Dryad Digital Repository: https://doi.org/10.5061/dryad.m63xsj3xj [33].

Authors' contributions. Q.-W.Z. and C.-L.L. conceived the project and set up the multiphysical models and designed the experiments; Q.-W.Z. measured and analysed the electrochemical data from simulation models; Q.-W.Z. wrote the manuscript; C.-L.L. coordinated the study and helped draft the manuscript; C.-L.L. Z.S. and J.-G.Y. reviewed and edited the manuscript. All authors gave the final approval for publication.

Competing interests. The authors declare no competing interests.

Funding. We acknowledge the financial support of the National Natural Science Foundation of China (grant 51504099 and grant U1407202).

Acknowledgements. We thank Jun-Xiang Ye for the instructions of Origin.

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
