## [Reviewer comments · Royal Society Open Science]

Review History

RSOS-191124.R0 (Original submission)

Review form: Reviewer 1

Is the manuscript scientifically sound in its present form?

No

Are the interpretations and conclusions justified by the results?

Yes

Is the language acceptable?

Yes

Do you have any ethical concerns with this paper?

No

Have you any concerns about statistical analyses in this paper?

No

Recommendation?

Reject

Comments to the Author(s)

In this manuscript, authors reported a modeling study of lithium cell by coupling electric field, flow field and concentration distribution together. They developed a 'multiphase model' using COMSOL software package and conducted optimization calculation to study the lithium and chlorine bubble production. Overall, this is an interesting work, but there are still many places that might need to be further improved and clarified:

1. Since 'lithium electrolysis process' was considered in the model, it is not clear what electrochemical reactions and how they were applied on the electrolyte and electrodes interface in their model.
2. In the '3.2.1 electric field' section, what kind boundary conditions were applied on two electrode surface? and How to couple them with the given equation-5? Besides, it might be not appropriate to use 'equation-5' for both solid electrode and liquid electrolyte since the physical meaning of 'sigma' in them will be different.
3. In section '3.2.2 velocity field', parameters including subscripts in given equations should be well explained (e.g. eq-6,8,9,10,11), otherwise, it is very difficult for readers to understand their meanings.
4. In section '3.2.3', 'R_i is equal to the electrolysis reaction generation rate when on the electrodes...', Question is what electrolysis reaction generation rate was used on the electrode surface?
5. In figure 2, it is very confusing which part is electrode, and which part stands for electrolyte, can authors clarify and mark them on the figure?
6. In their Table 3, it is not clear how 'electrochemical reactions' were applied as boundary conditions in the 'Concentration field'.
7. In addition, there are a few typos and grammar mistakes that need to be removed.

Review form: Reviewer 2**Is the manuscript scientifically sound in its present form?**

Yes

Are the interpretations and conclusions justified by the results?

Yes

Is the language acceptable?

Yes

Do you have any ethical concerns with this paper?

No

Have you any concerns about statistical analyses in this paper?

No

Recommendation?

Accept with minor revision (please list in comments)

Comments to the Author(s)

This paper studies the electrolysis efficiency of lithium cell based on electro-chemical and multiphase model. The authors analyzed a series of energy features of the as-studied system. This is an interesting study, which well-suits for the journal. The results are of good novelty and significance. The manuscript is well-prepared. Before it is acceptable, I have several concerns that should be carefully addressed by the authors.

Therefore, I recommend that this nice piece of work is acceptable for Royal Society Open Science after a minor revision. Details of my comments are shown as follows.

1. In the introduction the authors may refer to the aluminum reduction cell.
2. The structure of the model was shown before the structured mesh in Figures 2 may be clearer for this article.
3. Line 127, the "starting conditions" is "initial conditions".
4. There are several typos in this manuscript. Please double-check the paper.
5. The author should edit formula alignment and make them more beautiful, such equation 1-5, and 23-24.

Decision letter (RSOS-191124.R0)

16-Sep-2019

Dear Dr Liu:

Title: Analysing and Optimising the Electrolysis Efficiency of a Lithium Cell Based on Electro-Chemical and Multiphase Model

Manuscript ID: RSOS-191124

The editor assigned to your manuscript has now received comments from reviewers. We would like you to revise your paper in accordance with the referee and Subject Editor suggestions which can be found below (not including confidential reports to the Editor). Please note this decision does not guarantee eventual acceptance.

Please submit your revised paper before 09-Oct-2019. Please note that the revision deadline will expire at 00.00am on this date. If we do not hear from you within this time then it will be assumed that the paper has been withdrawn. In exceptional circumstances, extensions may be possible if agreed with the Editorial Office in advance. We do not allow multiple rounds of revision so we urge you to make every effort to fully address all of the comments at this stage. If deemed necessary by the Editors, your manuscript will be sent back to one or more of the original reviewers for assessment. If the original reviewers are not available we may invite new reviewers.

When submitting your revised manuscript, you must respond to the comments made by the referees and upload a file "Response to Referees" in "Section 6 - File Upload". Please use this to

document how you have responded to the comments, and the adjustments you have made. In order to expedite the processing of the revised manuscript, please be as specific as possible in your response.

On behalf of the Subject Editor Professor Anthony Stace and the Associate Editor Professor Kim Jelfs.

RSC Associate Editor:
Comments to the Author:
(There are no comments.)

RSC Scientific Editor:
Comments to the Author:
(There are no comments.)

Reviewers' Comments to Author:
Reviewer: 1

Comments to the Author(s)

In this manuscript, authors reported a modeling study of lithium cell by coupling electric field, flow field and concentration distribution together. They developed a 'multiphase model' using COMSOL software package and conducted optimization calculation to study the lithium and chlorine bubble production. Overall, this is an interesting work, but there are still many places that might need to be further improved and clarified:

1. Since 'lithium electrolysis process' was considered in the model, it is not clear what electrochemical reactions and how they were applied on the electrolyte and electrodes interface in their model.
2. In the '3.2.1 electric field' section, what kind boundary conditions were applied on two electrode surface? and How to couple them with the given equation-5? Besides, it might be not appropriate to use 'equation-5' for both solid electrode and liquid electrolyte since the physical meaning of 'sigma' in them will be different.
3. In section '3.2.2 velocity field', parameters including subscripts in given equations should be well explained (e.g. eq-6,8,9,10,11), otherwise, it is very difficult for readers to understand their meanings.

4. In section '3.2.3', 'Ri is equal to the electrolysis reaction generation rate when on the electrodes...', Question is what electrolysis reaction generation rate was used on the electrode surface?
5. In figure 2, it is very confusing which part is electrode, and which part stands for electrolyte, can authors clarify and mark them on the figure?
6. In their Table 3, it is not clear how 'electrochemical reactions' were applied as boundary conditions in the 'Concentration field'.
7. In addition, there are a few typos and grammar mistakes that need to be removed.

Reviewer: 2

Comments to the Author(s)

This paper studies the electrolysis efficiency of lithium cell based on electro-chemical and multiphase model. The authors analyzed a series of energy features of the as-studied system. This is an interesting study, which well-suits for the journal. The results are of good novelty and significance. The manuscript is well-prepared. Before it is acceptable, I have several concerns that should be carefully addressed by the authors.

Therefore, I recommend that this nice piece of work is acceptable for Royal Society Open Science after a minor revision. Details of my comments are shown as follows.

1. In the introduction the authors may refer to the aluminum reduction cell.
2. The structure of the model was shown before the structured mesh in Figures 2 may be clearer for this article.
3. Line 127, the "starting conditions" is "initial conditions".
4. There are several typos in this manuscript. Please double-check the paper.
5. The author should edit formula alignment and make them more beautiful, such equation 1-5, and 23-24.

Author's Response to Decision Letter for (RSOS-191124.R0)

See Appendix A.

RSOS-191124.R1 (Revision)

Review form: Reviewer 1

Is the manuscript scientifically sound in its present form?

Yes

Are the interpretations and conclusions justified by the results?

Yes

Is the language acceptable?

Yes

Do you have any ethical concerns with this paper?

No

Have you any concerns about statistical analyses in this paper?

Yes

Recommendation?

Accept as is

Comments to the Author(s)

Authors have addressed my concern adequately, and made necessary changes accordingly. I think this version can be considered for publication

Decision letter (RSOS-191124.R1)

19-Nov-2019

Dear Dr Liu:

Title: Analysing and Optimising the Electrolysis Efficiency of a Lithium Cell Based on Electro-Chemical and Multiphase Model

Manuscript ID: RSOS-191124.R1

It is a pleasure to accept your manuscript in its current form for publication in Royal Society Open Science. The chemistry content of Royal Society Open Science is published in collaboration with the Royal Society of Chemistry.

On behalf of the Subject Editor Professor Anthony Stace and the Associate Editor Professor Kim Jelfs.

RSC Associate Editor:
Comments to the Author:
(There are no comments.)

RSC Subject Editor:
Comments to the Author:
(There are no comments.)

Reviewer(s)' Comments to Author:
Reviewer: 1

Comments to the Author(s)
Authors have addressed my concern adequately, and made necessary changes accordingly. I think this version can be considered for publication

Appendix A

Detailed response to the reviewers of manuscript RSOS-191124 Analysing and Optimising the Electrolysis Efficiency of a Lithium Cell Based on Electro-Chemical and Multiphase Model

By Qian-Wen Zhao, Cheng-Lin Liu, Ze Sun, Jian-Guo Yu

Dear Dr. Laura Smith and Reviewers

Thanks for your letter with the comments and suggestions concerning our manuscript entitled “Analysing and Optimising the Electrolysis Efficiency of Lithium Cell Based on Electro-Chemical and Multiphase Model” (ID: RSOS-191124). Those comments are all valuable and very helpful for revising and improving our manuscript. Our manuscript has been carefully revised based on the reviewer’s comments, and the major revisions were marked in red. The detail replies to the reviewer’s comments and suggestions were listed as below.

Reviewer 1

Comment 1

Since ‘lithium electrolysis process’ was considered in the model, it is not clear what electrochemical reactions and how they were applied on the electrolyte and electrodes interface in their model.

Reply:

The reviewer’s advice is very valuable for our paper. The authors modified the section of introduction and the boundary conditions in the revised MS.

Action Taken:

In the section of introduction,

For the second method, metallic lithium is typically produced by the electrolysis of LiCl, while the raw material results from the ore or brine. The molten salt LiCl-KCl (42:58 in mass ratio) with a low eutectic point of 625.15 K has been adapted for electrolysis. These phenomena at each electrode are running according to the following electrochemical reactions:

In the section of the boundary conditions,

Table 3. Boundary Conditions in this Model

Boundary	Velocity field	Electric field	Concentration field
Anode (contact with electrolyte)	Gas inlet, slip for liquid	5 kA	$2Cl^- - 2e^- \rightarrow Cl_{2(gas)}$
Cathode (contact with electrolyte)	No slip for liquid	0 V	$Li^+ + e^- \rightarrow Li_{(liquid)}$
Outlet	Slip for liquid, gas outlet	Electrical insulation	Gas outlet
Side-wall	No slip for liquid	Electrical insulation	Wall
bottom	No slip for liquid	Electrical insulation	Wall

Comment 2

In the ‘3.2.1 electric field’ section, what kind boundary conditions were applied on two electrode surface? and How to couple them with the given equation-5? Besides, it might be not appropriate to use ‘equation-5’ for both solid electrode and liquid electrolyte since the physical meaning of ‘sigma’ in them will be different.

Reply:

The reviewer’s advice is correct and very helpful for the article. The ‘sigma’ in the electrode and electrolyte is different. However, in these model, the ‘sigma’ only apply in the electrolyte. The boundary conditions of electric field were applied on the two electrode surface where contact with the electrolyte. The amendment is shown in the Action Taken part.

Action Taken:

In the section of boundary conditions

The starting conditions for the electrolysis cell are listed in Table 2.

Table 2. Starting Conditions in the Model

Transfer process	Description
Electrical Potential	Cell voltage equals 0 V
Mass Transfer	Reactive species concentration equals $19.2 \text{ kmol}\cdot\text{m}^{-3}$
Velocity field	Velocity equals zero

The boundary conditions used in this model are listed in Table 3.

Table 3. Boundary Conditions in this Model

Boundary	Velocity field	Electric field	Concentration field
----------	----------------	----------------	---------------------

Anode (contact with electrolyte)	Gas inlet, slip for liquid	5 kA	$2Cl^- - 2e^- \rightarrow Cl_{2(gas)}$
Cathode (contact with electrolyte)	No slip for liquid	0 V	$Li^+ + e^- \rightarrow Li_{(liquid)}$
Outlet	Slip for liquid, gas outlet	Electrical insulation	Gas outlet
Side-wall	No slip for liquid	Electrical insulation	Wall
bottom	No slip for liquid	Electrical insulation	Wall

Comment 3

In section ‘3.2.2 velocity field’, parameters including subscripts in given equations should be well explained (e.g. eq-6,8,9,10,11), otherwise, it is very difficult for readers to understand their meanings.

Reply:

Thanks very much for the reviewer’s advice. The authors revised the manuscript again and modified the section of 3.2.2 in the revised manuscript as shown in the Action Taken part.

Action Taken:

In the 3.2.2,

In this equation, u is the speed ($m \cdot s^{-1}$), p represents the pressure (Pa), ρ is the fluid density ($kg \cdot m^{-3}$), g is the gravitational acceleration ($m \cdot s^{-2}$), and $\mu_{eff,k}$ is the effective viscosity.

In this equation, F_{added} is the added force on the fluid ($N \cdot m^{-3}$), and T_k^{turb} is the turbulence stress ($N \cdot m^{-3}$).

Comment 4

In section ‘3.2.3’, ‘ Ri is equal to the electrolysis reaction generation rate when on the electrodes...’, Question is what electrolysis reaction generation rate was used on the electrode surface?

Reply:

The reviewer’s advice is very valuable for our paper. The description in the original article is wrong. Ri was set to zero in the mathematical model. The authors modified in the revised MS in the Action Taken.

Action Taken:

In the section of 2.2.3,

where R_i means the homogeneous reaction rate of species i in the electrolyte. In the cell, R_i is equal to the electrolysis reaction generation rate when on the electrodes and equals zero.

Comment 5

In figure 2, it is very confusing which part is electrode, and which part stands for electrolyte, can authors clarify and mark them on the figure?

Reply:

The reviewer's advice is very valuable for our paper. The authors modified the Figure 2 in the revised MS in the Action Taken.

Action Taken:

Fig. 2. Structured and Mesh of the model

Comment 6

In their Table 3, it is not clear how ‘electrochemical reactions’ were applied as boundary conditions in the ‘Concentration field’.

Reply:

The reviewer’s advice is very valuable for our paper. The authors modified the Table 3 in the revised MS in the Action Taken.

Action Taken:

Table 3. Boundary Conditions in this Model

Boundary	Velocity field	Electric field	Concentration field
Anode (contact with electrolyte)	Gas inlet, slip for liquid	5 kA	$2Cl^{-} - 2e^{-} \rightarrow Cl_{2(gas)}$
Cathode (contact with electrolyte)	No slip for liquid	0 V	$Li^{+} + e^{-} \rightarrow Li_{(liquid)}$
Outlet	Slip for liquid, gas outlet	Electrical insulation	Gas outlet
Side-wall	No slip for liquid	Electrical insulation	Wall
bottom	No slip for liquid	Electrical insulation	Wall

Comment 7

In addition, there are a few typos and grammar mistakes that need to be removed.

Reply:

Thank you very much for the careful check of the article. The authors have checked the whole manuscript carefully and the typos are revised in the manuscript as shown in Action Taken.

Finally, we appreciate very much for your time in editing our manuscript and the referees for their valuable suggestions and comments. I am looking forward to hearing your final decision when it is made.

Reviewer 2

Comment 1

In the introduction the authors may refer to the aluminum reduction cell.

Reply:

Thank you very much for the comment. And the comment is crucial for achieving satisfying result.

Action Taken:

In the introduction section,

In the aluminum reduction cell, Tessier et al.[17] developed a multiblock partial least squares modeling approach for multivariate analysis and monitoring of aluminum reduction smelters and other electrochemical processes, and Zhang et al. [18] developed a microscale modeling approach for investigation of bubble dynamics in the aluminum smelting process. Zhan et al.[19] used the a 3D CFD PBM model to analyzed the effects of different cell design and operating parameters on the gas–liquid two-phase flows and bubble distribution characteristics under the anode bottom regions in aluminum electrolysis cells.

- 9 Zhan S, Wang Z, Yang J, Zhao R, Li C, Wang J, Zhou J. 2017 3D Numerical Simulations of Gas–Liquid Two-Phase Flows in Aluminum Electrolysis Cells with the Coupled Model of Computational Fluid Dynamics–Population Balance Model. *Ind. Eng. Chem. Res.* 56, 8649–8662. (doi:10.1021/acs.iecr.7b01765)
- 17 Tessier J, Duchesne C, Tarcy GP, Gauthier C, Dufour G. 2012 Multivariate Analysis and Monitoring of the Performance of Aluminum Reduction Cells. *Ind. Eng. Chem. Res.* 51, 1311–1323. (doi:10.1021/ie201258b)
18. Zhang K, Feng Y, Schwarz P, Wang Z, Cooksey M. 2013 Computational Fluid Dynamics (CFD) Modeling of Bubble Dynamics in the Aluminum Smelting Process. *Ind. Eng. Chem. Res.* 52, 11378–11390. (doi:10.1021/ie303464a)
19. Zhan S, Wang J, Wang Z, Yang J. 2018 Computational Fluid Dynamics-Population Balance Model Simulation of Effects of Cell Design and Operating Parameters on Gas–Liquid Two-Phase Flows and Bubble Distribution Characteristics in Aluminum Electrolysis Cells. *Jom* 70, 229–236. (doi:10.1007/s11837-017-2636-8)

Comment 2

The structure of the model was shown before the structured mesh in Figures 2 may be clearer for this article.

Reply:

Thank you very much for the comment and the comment is crucial for improving our paper. The authors modified the Figure 2 with adding the structure of the physical model in the revised MS in the Action Taken.

Action Taken:

Fig. 2. Structured and Mesh of the model

Comment 3

Line 127, the “starting conditions” is “initial conditions”.

Reply:

Thank you for pointing out the problem.

Action Taken:

In section of 3.3, the title was modified as “Initial and Boundary Conditions”. The table was modified as “Table 2. Initial condition in the model”

Comment 4

There are several typos in this manuscript. Please double-check the paper.

Reply:

Thank you very much for the careful check of the article. The authors have checked the whole manuscript carefully and the typos are revised in the manuscript as shown in Action Taken.

Comment 5

The author should edit formula alignment and make them more beautiful, such equation 1-5, and 23-24.

Reply:

Thank you for the comment.

Action Taken:

In page 3,

In this instance, the electric field could be described by the Ohm's law, continuity law and the Gauss's law as below:

$$J = \sigma E \quad (1)$$

$$\frac{\partial q}{\partial t} + \nabla \cdot J = 0 \quad (2)$$

$$E = -\nabla \varphi \quad (3)$$

where J ($A \cdot m^{-2}$) is the current density; σ ($S \cdot m^{-1}$) is the electric conductivity; E ($V \cdot m^{-1}$) is the electric field; q ($C \cdot m^{-3}$) represents the charge density; and φ (V) is the electric potential.

Connecting the above equations, we obtain the following:

$$\nabla \cdot (\sigma \nabla \varphi) = \frac{\partial q}{\partial t} \quad (4)$$

During the electrolysis process in the cell, the electric field stays stable and there is no other charge produced. Equation (4) can be simplified as

$$\nabla \cdot (\sigma \nabla \varphi) = 0 \quad (5)$$

In page 7,

The electric reaction relating to current on the electrode surface is as follows:

$$\sum_{ox} \nu_{ox} S_{ox} + ne \leftrightarrow \sum_{red} \nu_{red} S_{red} \quad (23)$$

The reactant S_{ox} on this electrode–electrolyte interface gets electrons and is reduced to the product S_{red} . ν_{ox} is the stoichiometric number. Those species are the only mass flowing out the reaction surface. The species flow can be calculated by Faraday Law:

$$R_i = \sum_m \frac{\nu_{i,m} i_l}{nF} \quad (24)$$

i_l is the local current density ($\text{A}\cdot\text{m}^{-2}$) and n is the reacting electron number.